# Preliminary Research on the Effect of Hyperbaric Oxygen Therapy in Patients with Post-COVID-19 Syndrome

**DOI:** 10.3390/jcm12010308

**Published:** 2022-12-30

**Authors:** Diana Kitala, Wojciech Łabuś, Jerzy Kozielski, Przemysław Strzelec, Mariusz Nowak, Grzegorz Knefel, Piotr Dyjas, Krzysztof Materniak, Jolanta Kosmala, Jolanta Pająk, Jolanta Czop, Barbara Janda-Kalus, Barbara Marona, Anna Nowak-Wróżyna, Marcin Gierek, Jan Szczegielniak, Marek Kucharzewski

**Affiliations:** 1Medical Research Agency, Stanisława Moniuszki 1a St., 00-014 Warsaw, Poland; 2Dr Stanisław Sakiel Center for Burns Treatment, Jana Pawła II 2, 41-100 Siemianowice Śląskie, Poland; 3Faculty of Physical Education and Physiotherapy, Opole University of Technology, Prószkowska 76, 45-758 Opole, Poland

**Keywords:** SARS-CoV-2, post-COVID syndrome, hyperbaric oxygen therapy HBOT

## Abstract

Negative consequences and medical complications of COVID-19 can persist for up to several months after initial recovery. These consequences can include stroke, diabetes, decreased lung diffusing capacity, sleep apnea, pulmonary fibrosis, arrhythmia, myocarditis, fatigue, headaches, muscle aches, heart rate fluctuations, sleep problems, memory problems, nervousness, anxiety, and other neurological disorders. Thirty-one patients who reported symptoms related to previous COVID-19 disease of both sexes were enrolled in the initial program. The patients underwent compression sessions in a multiplace hyperbaric chamber. Each patient underwent a cycle of 15 compressions. Before the first session, each participant completed a venous blood gas test, a Fullerton test, and two spirometry tests (one before the Fullerton test and one after the test). Patients completed psychotechnical tests, a questionnaire on quality of life (Polish version of EQ-5D-5L), and a questionnaire on specific symptoms accompanying the disease and post-infection symptoms. The results showed significant improvements in areas such as quality of life, endurance and strength, some spirometric parameters, the anion gap and lactate levels, working memory, and attention in the group of treated patients. In contrast, there were no changes in pH, pO2, pCO2, glucose, and excess alkaline values. A follow-up interview confirmed that the beneficial effects were maintained over time. Considering the results obtained, including the apparent improvement in the patient’s clinical condition, it can be concluded that the use of 15 compression sessions was temporarily associated with a noticeable improvement in health and performance parameters as well as improvement in certain blood gas parameters.

## 1. Introduction

The COVID-19 coronavirus disease, caused by the SARS-CoV-2 virus, is associated with more than 50 different negative consequences and medical complications that can persist for weeks to months after initial recovery. Long-term consequences of COVID-19 infection can include stroke, diabetes, decreased lung diffusion capacity, sleep apnea, pulmonary fibrosis, cardiac arrhythmias, and myocarditis, among others [1,2,3,4,5,6]. In addition, patients complain for fatigue, headaches, muscle pain, heart rate fluctuations, sleep problems, memory problems, nervousness, anxiety, and other neurological disorders. Not only is SARS-CoV-2 virus infection fraught with serious health consequences, it is also associated with severe social stigma [3,4,5,6].

Post-COVID syndrome is a new clinical entity in the context of SARS-CoV-2 infection. Its symptoms persist for more than three weeks after the diagnosis of COVID-19, and its incidence rate ranges from 10% to 85%. Prolonged inflammation plays a key role in the pathogenesis of this syndrome and can cause neurological complications, cognitive impairment [3], fever, sore throat, cough, shortness of breath, and chest pain [4]. The complications mentioned above have been observed in mildly symptomatic or even asymptomatic patients [5]. In some cases, fatigue has even been noted 12 weeks after discharge from the hospital. Chest pain and shortness of breath may be associated with pulmonary embolism [4]. Previously, patients with mild symptoms of the disease showed significant changes in chest X-ray images, such as a “frosted glass” image [6]. To overcome these problems, anti-inflammatory treatment should be used. A treatment with documented anti-inflammatory effects is hyperbaric oxygen therapy (HBO), which has been used for almost a century [7]. HBO has been described as an effective therapy for the treatment of asthma [8] and numerous other autoimmune diseases such as atopic dermatitis [9,10].

HBO therapy (HBOT) is also a well-established method of combating hypoxia, including in patients suffering from COVID-19 [8,9,10]. At the time of writing this report, there were several registered clinical trials on this topic, such as: “Hyperbaric Oxygen Therapy in Post-COVID-19 Syndrome—HBOTpCOVID” and “Hyperbaric Oxygen Therapy in Treatment of LongCOVID—HOT-LoCO Syndrome.” However, despite a need to treat post-COVID syndrome, only one study was actively recruiting participants. This is especially notable since, as clinical trials have shown, HBO can alleviate oxygen debt and reduce inflammatory responses. However, the mechanism by which HBO treatment reduces inflammation is still not fully understood [11,12].

The positive preliminary results of clinical trials on the use of HBO in post-COVID discussed above may have been confirmed by a paper published by the team of Robbins et al. [13] who described satisfactory results of HBO therapy in a group of 10 patients suffering from post-COVID syndrome. The study showed statistically significant improvement in all patients studied after a series of 10 compressions on HBO [13]. In addition, one case of a 55-year-old Caucasian male suffering from complaints of post-COVID syndrome three months after recovery was described. The HBO therapy used in this case also resulted in a significant improvement in the parameters studied [14]. There have also been reports of a positive effect of HBO in patients with severe respiratory failure who were still suffering from COVID-19 infection [11,12,15,16,17].

It should be further emphasized that patients with the so-called long-COVID/post-COVID syndrome are often socially or occupationally excluded due to their symptoms and struggle not only with the disease itself, but also with depression and long-term health and occupational consequences. Therefore, there is an urgent need to implement an effective therapeutic program that is able to counteract the negative symptoms of the long COVID/post-COVID syndrome. In a view of the literature reports on the positive results of HBOT in post-COVID therapy, we decided to conduct a preliminary study of the effectiveness of HBOT in post-COVID patients visiting our center. Thus, the presented program was primarily designed as a desperate attempt to help suffering patients using a recognized and widely used therapeutic method (HBOT) in a novel application.

## 2. Materials and Methods

The procedures followed were in accordance with the ethical standards of the Helsinki Declaration (1964, amended most recently in 2008), which was created by the World Medical Association. The scientific protocol presented in this study received a positive opinion from the Bioethics Committee of the Medical University of Silesia in Katowice, Poland, number PCN/0022/KB1/38/21 of 20 April 2021.

The study lasted from 10 May 2021 to 10 November 2021.

### 2.1. Characteristics of Patients

31 patients (out of 32 patients) who reported symptoms related to a previous infection of COVID-19 of both sexes were enrolled in the study program. One patient was excluded from the program due to emphysema. Patients were informed about the purpose and nature of the experiment in a clear and accessible manner. Patients were asked about their vaccination status. Professional, technical language was avoided. Each patient was informed of the possible side effects of the therapy and the possibility of withdrawing from the experiment at any time. Each patient gave their informed consent to participate in the experiment.

#### 2.1.1. Inclusion Criteria

-giving informed consent to participate in the study;-symptoms of fatigue, weakness, bone and muscle pain, joint pain, problems with performing a full breath, problems with concentration, memory problems, headache, nervousness, or sleep problems for more than three weeks after the resolution of COVID-19.

#### 2.1.2. Exclusion Criteria

-refusing to give informed consent to participate in the study;-pneumothorax, emphysema, or chronic obstructive pulmonary disease (COPD);-claustrophobia;-epilepsy or schizophrenia.

### 2.2. Rehabilitation in a Hyperbaric Chamber

Patients underwent compression sessions in a HAUX-STARMED 2500 multiplace hyperbaric chamber (HAUX-LIFE-SUPPORT). Each patient underwent a cycle of 15 compressions. Compression sessions were held 5 times a week, with a weekend break (3 series of 5 compressions were performed). Each session lasted 1 h and 15 min for a total time of 18 h and 45 min per patient. Compressions were conducted at a pressure of 2.5 ATA (1.5 bar) using medical oxygen. Patients breathed 100% oxygen.

The pressure inside the hyperbaric chamber was greater than sea level pressure, which is defined as 1 atmosphere absolute (1 ATA).

### 2.3. Patient Tests

Before the first session, each participant underwent a venous blood gas test, a Fullerton test, and two spirometry tests (one before the Fullerton test and one after the test). The venous blood gas test was performed on a RADIOMETER ABL 800 FLEX (Ra-diometer). To reduce measurement error, spirometry was performed on a Spirolab III + SpO2 (MIR) apparatus by one person. The Fullerton functional fitness test was chosen because it is uniquely suited for the multidimensional assessment of physical fitness, including in people with chronic obstructive pulmonary disease, and is therefore recommended for use by the International Council of Sport and Physical Education Sciences (ICSSPE-CIEPSS) [18]. The full set of tests was repeated after 5, 10, and 15 sessions to assess the effect of the chamber on patient fitness. The Fullerton test was conducted to evaluate each patient’s strength, aerobic endurance, flexibility, and agility/dynamic balance. The full test consists of 6 subsections designed to assess physiological parameters related to lower and upper body strength, aerobic endurance, lower and upper body flexibility, and agility/dynamic balance [19]. The Fullerton test included a left- and right-hand dynamometer test (newtons), a back scratch test (cm), a chair stand test (no. of stands), an 8-foot walk and return test, a two-minute step test (no. of steps), and a sit and reach test (cm).

A dynamometer test (grip strength test) was used to verify upper body strength. To perform the dynamometer test (grip strength test), each patient held a handheld hydraulic dynamometer (Jamar, Poland) and squeezed it with their hand. The exercise was performed twice for each hand.

A “back scratch” test was used to test upper body flexibility. To perform the back scratch test, each patient tried to bring their hands together as close as possible behind their backs. One hand reached behind the shoulder and the other reached the middle of the back. The number of centimeters between the extended middle fingers (+ or −) was measured with a sterile ruler. Flexibility was checked at the shoulder joint and shoulder arch on both the right and left sides.

The 30 s chair stand test was used to verify lower body strength. To perform the 30 s chair stand test, each patient made continuous attempts to stand up from a chair for thirty seconds. The number of full lunges completed with the patient’s arms folded over their chest in thirty seconds was counted.

To verify lower body flexibility, a sit and reach test was performed. To perform the sit and reach test, each patient sat on the edge of a chair with their legs extended and attempted to touch their feet by reaching their hands toward their toes. The number of centimeters (+ or −) from their outstretched fingers to the tip of their toes was measured.

To verify agility/dynamic balance, an 8-foot walk and return test was conducted. To perform the 8-foot walk and return test, each patient attempted to stand up from a seated position, walk backwards 8 feet (243.8 cm), and return to a seated position on a chair. The number of seconds required to walk the required distance was measured.

To verify aerobic endurance, a two-minute step test was conducted. To perform the two-minute step test, each patient attempted to walk in place by lifting each knee to a point halfway between the patella and the iliac crest over a two-minute period. The number of complete steps taken in two minutes was measured.

Pulse oximeter and spirometry tests were performed before and after the Fullerton test.

In addition, psychotechnical tests were performed during each session using a device with RehaCom software (Hasomed GmbH, Magdeburg, Germany). Patients were assessed using tests of divided attention and working memory. This was followed by training using divisible attention, attention, and concentration programs.

During each session, patients also filled out a questionnaire on quality of life (Polish version of the EQ-5D-5L) and a questionnaire on the specific symptoms that accompanied the disease and the symptoms they experienced after the infection.

As the EQ-5D-5L survey is a standardized document of international significance, it did not require validation prior to use. The survey consisted of five questions covering areas of daily life such as mobility, self-care, usual activities (e.g., work, study, etc.), pain/discomfort, and anxiety/depression. Five graded responses were prepared for each question, starting with a response stating that the patient had no problems with the question asked and ending with responses regarding their inability to perform a given activity. In addition, the questionnaire included a quality of life scale. The scale ranged from 0 to 100, with 0 indicating the worst imaginable state of health and 100 indicating the best imaginable state of health. Patients filled out a new questionnaire before their first com-pression and after 5, 10, and 15 compressions. In total, each patient completed four questionnaire sheets.

A questionnaire on the specific symptoms that accompanied the disease and the post-infection symptoms was specifically designed for the proposed program. The questionnaire consisted of questions about headaches, muscle pain, joint pain, fatigue, concentration problems, memory problems, nervousness, olfactory problems, taste problems, sleep problems, feelings of anxiety, breathing problems, heart rate fluctuations or hypertension, gastrointestinal symptoms (e.g., abdominal pain, diarrhea), and glycemic problems such as diabetes. The patients’ responses to questions were graded on a 4-point scale (very rarely, rarely, often, very often). Patients gave subjective answers according to their feelings. The questions asked about the patients’ feelings both before and during the infection and about symptoms that persist now. Patients answered the questions after every 5 sessions of HBO compressions (i.e., after 5, 10, and 15 sessions). Due to the continuous nature of the study, it was not possible to validate the proposed questionnaire. Patients completed the same survey sheet before the first compression (questions about how they felt before the infection, during the infection and now—after the infection), and after 5, 10, and 15 compressions.

### 2.4. Long-Term Evaluation

A telephone survey was conducted with participants 2–24 weeks after program completion to verify the long-term impact of HBO on their quality of life. Patients were asked whether the positive effect that HBO therapy had on their health status continued after the program was completed. Patients were asked to rate their quality of life on a 100-point scale; they were also asked about what area of their health they felt improved the most and what area they felt improved the least.

### 2.5. Statistical Analysis

All tests were performed using the STATISTICA 12 software (StatSoft, Dell Inc., Round Rock, TX, USA). The distribution of each variable was analyzed using the Shapiro–Wilk test and the equality of variance was checked using the Levene’s test. For variables that were measured at multiple time points, an ANOVA with repeated measures was used. A multivariate regression analysis was performed to assess the impact of sociodemographic parameters on quality of life before and after therapy. A Chi-square test was used for dichotomous data. The significance level was set at 0.05 (5%).

## 3. Results

The experiment began in May 2021 and was completed in November 2021. At that time, 31 patients had been recruited and examined (Table 1). The average age of the patients that were admitted to rehabilitation was 55 ± 9 years. 74% of the participants in the experiment had never smoked and 26% had smoked in the past. None of the participants reported actively smoking when they were infected with COVID-19 or during hyperbaric oxygen therapy. The mean BMI was 29.6 ± 4.4 (90% of subjects were overweight or obese). The average duration of post-COVID syndrome was 6.2 ± 3.5 months (in 35% of participants it was longer than 6 months). Overall, 48% of subjects were hospitalized for a COVID-19 infection. Hospitalization of more than 2 weeks was required in 29% of patients. None of the patients had been vaccinated prior to infection.

One patient did not qualify for the study for medical reasons.

### 3.1. Quality of Life

Patients initially declared their quality of life on the EQ-5D-5L questionnaire to be 67.8 ± 18.5 (mean ± SD on a scale of 0–100). A multivariate analysis conducted to evaluate the impact of sociodemographic parameters (age, BMI, time from diagnosis to inclusion in therapy, duration of hospitalization) on quality of life before and after therapy showed that being overweight was the only factor that had a direct impact on quality of life before therapy (*p* = 0.029), while none of the tested parameters had an impact on quality of life after therapy. Before the start of compression, women and men declared a mean ± SD quality of life of 66.8 ± 19 and 68.8 ± 18.1, respectively. Quality of life improved after 5 compressions, reaching a mean value of 71.2 ± 18.5 (Figure 1a). After 10 compressions, the mean quality of life was 74.6 ± 19.1, and after 15 sessions in the hyperbaric chamber, it was 83.6 ± 13.6. This indicates that 15 HBO treatments is the minimum number required to significantly improve quality of life. The maximum quality of life score obtained after 15 sessions was 100 (the maximum on the scale) and the most common value (mode) was 90/100 points. Sixteen participants declared a quality of life of at least 90 points. On average, patients’ quality of life improved by 18% (maximum increase of 40 points). Compared to the scores before HBOT, the reported changes in quality of life after 5, 10, and 15 HBOT sessions were statistically significant (*p* < 0.001). Age did not improve quality of life during the experiment (*p* = 0.14). BMI influenced the change in the patients’ perceptions of their quality of life (*p* = 0.003) (Figure 1b). There were differences between normal-weight patients and those in the obesity II stage (*p* < 0.001). Additionally, gender was a factor that influenced patients’ perception of change in quality of life (*p* = 0.008). Males had a higher mean life quality than females before oxygenation and after 5 sessions, but woman achieved a similar mean quality of life after 10 and 15 sessions. 

Initially, 55% of participants reported walking difficulties (mild, moderate or severe), which decreased to 29% after 15 chamber sessions (including none severe; Figure 1c). 

The change in declared walking problems was statistically significant (*p* < 0.01). Full self-care ability (washing and dressing) before hyperbaric oxygen therapy was declared by 77% of participants and by up to 94% of participants after a cycle of 15 compressions (*p* < 0.01) (Figure 1d).

Another question from the EQ-5D-5L survey concerned problems with performing usual activities (e.g., work, study, household activities, family activities, and leisure activities). Before the series of compressions, 58% of respondents declared problems with performing the activities mentioned above to a minor, moderate, or severe degree. After the series of 15 compressions, only 22% of subjects continued to report problems, but none of them were severe and only one was moderate, which was a statistically significant change (*p* < 0.01) (Figure 1e). 

Feelings of anxiety or depression (mild to severe) were present before the start of therapy in 71% of the participants in the experiment and in only 35% (including no one with severe feelings) after completing a full cycle of compressions. This was a statistically significant decrease (*p* < 0.01) (Figure 1f). There was also a statistically significant decrease in perceived pain (*p* = 0.027) (Figure 1g). BMI status, age, and gender had no effect on these parameters during the experiment.

Given the recurrent symptoms that are characteristic of post-COVID syndrome, a proprietary questionnaire was created that included the type and severity of the most common symptoms reported by patients who had a history of SARS-Cov-2 infection,. The results of the questionnaire showed a significant decrease in patients’ perceived levels of headache due to nervousness, fatigue, sleep problems, memory problems, muscle pain, joint pain, and problems with taking a full breath (all *p* < 0.001) (Appendix A), (Figure 2). The change in concentration problems was also significant (*p* = 0.002), as was the decrease in problems with blood pressure spikes and heart rate fluctuations (*p* = 0.033) (Figure 2g). 

### 3.2. Assessment of General Physical Condition, Including Efficiency

Efficiency, strength, and flexibility were tested with the Fullerton test. Before the start of the therapy, the mean saturation was about 97 (minimum saturation 93) before exercise and about 95 (minimum saturation 70) after an exercise session (including a two-minute step test). The mean difference was −2.03, which decreased successively with each compression to −0.35 after a cycle of 15 treatments. The minimum saturation after a full cycle of compressions was 92 before exercise and 90 after a series of exercises (Appendix A).

Post-exercise saturation changed significantly during the experiment (*p* = 0.032), while the difference between the saturation before the exercise and after exercise was not significant (Appendix A).

Endurance and strength tests clearly showed a positive effect of hyperbaric oxygen therapy on patient performance. Prior to the start of compression, four patients were unable to continue a two-minute step due to malaise and the average number of full lunges after getting up from the chair was about 14. The average time to complete the 8-foot (2.4 m) test, which involved getting up from the chair, walking away from and returning to the chair, and assuming a sitting position in the chair, was 6.7 s (Appendix A).

There was an improvement in performance as measured by the two-minute step test, with scores improving from about an average of 92 repetitions before compression to about an average of 126 repetitions after hyperbaric chamber compression. The change in this test score was statistically significant (*p* < 0.001). Gender and hospitalization for SARS-CoV-2 infection had no effect on this test outcome, while BMI affected the score. Post hoc analysis showed that normal weight subjects had a better score than overweight and obese subjects (*p* < 0.001).

The improvement in fitness was confirmed in the test that involved standing up from a chair in thirty seconds. The number of chair lifts obtained during the “standing up from a chair in thirty seconds” test changed significantly during the experiment. Gender and age did not affect this parameter, while BMI status did (similar to the two-minute step test). The number of seconds to perform the 8-foot test (standing up and walking) did not change significantly after successive compressions. Hospitalization for COVID-19 had no effect on this test.

However, significant changes were found in the sit up and reach test. Gender had no effect on this parameter, nor did the patient’s preexisting obesity or overweight. The improvement in flexibility was not reflected in the test involving touching the fingers of opposite hands behind the back (the back scratch test), but given that some patients had shoulder girdle problems, this result is not surprising (Appendix A).

There was a significant change in strength (in Newtons) in both the left and right hands, as measured with a dynamometer during treatment. Age, gender, and BMI were not factors in the improvement of strength in the left or right hand during the experiment. Whether less than or more than six months had elapsed between the onset of the disease and the start of the experiment also had no effect on the degree of strength improvement over time. The results of the dynamometry tests (left and right) were correlated with the results of the two-minute walk test (R = 0.38 and R = 0.36, respectively) (Appendix A).

### 3.3. Influence on the Parameters of Venous Blood Gasometry

There was a change in the anion gap (*p* = 0.0245) and lactate levels during treatment. This difference was not affected by age or BMI. The lactate value decreased from 1.35 ± 0.47 to 1.06 ± 0.4 (mean ± SD) after 15 compressions (Appendix A).

In contrast, there were no changes in pH (negative logarithm of hydrogen ion activity), pO2 (partial pressure of oxygen), pCO2 (partial pressure of carbon dioxide), or excess base. The observed changes in glucose levels were also not statistically significant. The pH obtained by gasometry before the start of compressions was negatively correlated with scores on the back scratch test (R = −0.46), the sit and reach test (R = −0.51), and the two-minute walk test (R = −0.56).

### 3.4. Assessment of Respiratory System Function

Before the start of the HBOT series, there was a negative correlation between the patient’s age and their spirometric FEV1 (forced expiratory volume in one second) (R = −0.4) and FEF2575 (forced mid-expiratory flow) (R = −0.4) values. There were no differences in changes in spirometric FEV1 (L) scores during hyperbaric oxygen therapy. However, there were significant differences in FVC (forced vital capacity) scores during treatment (higher in the younger group) between patients who were ≤55 years old compared to those who were >55 years old. A significant change was observed in the difference between FEV1 before exercise and FEV1 after exercise (*p* = 0.032; Appendix A). However, there were no differences in the changes of FEV1/FVC, PEF (peak expiratory flow), and FEF2575 values during hyperbaric chamber oxygen therapy. There were also no significant changes in the differences between FEV1/FVC, FEV1, PEF, and FEF2575 values measured before and after exercise. Before compression, the patients’ glucose levels were negatively associated with FEV/FVC (R = −0.49) and FEF2575 (R = −0.38) (Appendix A).

### 3.5. Assessment of Post-COVID Brain Fog and Cognitive Enhancement

The improvement in working memory was statistically significant (*p* = 0.015), as was the improvement in concentration and attention (*p* < 0.001) (Appendix A). 

### 3.6. Long Term Evaluation

We were able to obtain data on the long-term effects of COVID-19 infection from 30 out of 31 patients. Within this group, 90% of the subjects confirmed long-term improvement after hyperbaric oxygen therapy. Gender, age, and BMI status were not factors that contributed to the improvement obtained. The patients’ mean quality of life score after therapy (min. two weeks, max. six months) was 83.3 ± 12.8, which is a significant difference from the mean score of 67.8 ± 18.5 reported at baseline (*p* < 0.001). In addition, 47% of participants reported further improvements in physical performance, 43% had improved overall well-being and more energy, 6% had a resolution of muscle or joint pain, and the remainder reported improvements in headaches and sleep quality (Appendix A). Some individuals said they wanted to perform an additional series of compressions to improve their quality of life even further.

## 4. Discussion

The long-term consequences of COVID-19 infection have been described in patients with even mild symptoms and are still not well understood [4,5,20,21,22,23,24]. Post-COVID symptoms can occur even a year after initial recovery [25,26]. These results correspond directly with those obtained in the present study which found that more than one-third of the patients developed long COVID approximately six months after infection. In addition, all the patients included in the study complained of significantly reduced quality of life, which directly relates to the data reported in the literature. Patients often complained of having problems returning to routine activities of daily living [4,13,14,26]. In this specific approach, it should be noted that the interval between the patients’ recovery from COVID-19 and the start of HBOT lasted an average of six months. Thus, a reliable evaluation after such a long time may be a critical aspect of the present study. The explanation for this phenomenon, however, is mundane. Before qualifying for HBOT, all the patients underwent pulmonary rehabilitation and recovery immediately after recovering from COVID-19. However, this treatment did not lead to any significant improvement. Therefore, had it not been for the novel application of HBOT in this area, the post-COVID period in this group of patients could have lasted much longer than it did. In addition, although the phenomenon of long-term persistence of post-COVID symptoms has been confirmed in the literature, and it is natural that patients may experience persistent symptoms for much longer than six months, the analysis of such retrospective data may raise some controversy. 

Indeed, when it came to conducting a proper clinical trial, this type of data could not be relied upon. However, the intention of the authors of this paper was not to conduct a clinical trial, but rather to make a desperate attempt to help seriously ill patients whose conditions (in some cases) disqualified them from working. In addition, it can be noted that many other authors, such as Robbins et al. [13], used retrospective evaluation in their studies. The research team of Robbins et al. (2021) demonstrated the effectiveness of HBO therapy in patients with fatigue symptoms after COVID-19 infection [27]. Robbins et al. used HBO because of its effect on chronic fatigue syndrome. We confirmed that a large percentage of patients suffered from fatigue during long COVID, which may explain the mechanism underlying the effect of HBO effect on quality of life. In our study, HBO therapy improved quality of life as measured by the international standardized EQ-5D-5L questionnaire from an average of 67.8 points (0–100 point scale) to an average of 83.6 points. This improvement even continued about six months after the end of therapy (mean quality of life score of 83.3 two weeks–six months after the end of therapy). A complete ability to engage in self-care behaviors (washing and dressing) before hyperbaric oxygen therapy was declared by 77% of participants and by up to 94% of participants after a cycle of 15 compressions (*p* < 0.01). Before HBO therapy, 71% of the participants in the experiment reported feeling anxious or depressed (mild to severe), while only 35% reported such feelings after completing a full cycle of compressions (none of which were severe) (*p* < 0.01). This was a statistically significant decrease. Feelings of fatigue and nervousness also decreased. Even after about six months following the completion of therapy, patients continued to report improvements in overall well-being and sleep quality. The results presented here are consistent with those obtained by the research team of Robbins et al. [13]. 

An interesting aspect of this work may be the fact that the reported level of quality of life before COVID-19 was surprisingly low in the group of patients with normal weight (i.e., patients who were not overweight or obese). It is possible that the small sample size and resulting high bias could be a rational explanation for this finding.

However, in the present study, an additional follow-up study was performed. Headaches and musculoarticular pains did not return for a certain amount of time after the end of HBOT (Figure 2). Given this finding, it can be noted that a properly performed follow-up examination should take place approximately 6–12 months after the completion of therapy. Thus, our lack of a long-term follow-up can be considered to be one of the limitations of this work. However, it can be noted that the results obtained in our study nevertheless allowed us to propose conclusions regarding, for example, the possibility of performing prophylactic HBOT sessions in post-COVID patients. Moreover, the feeling of having a problem with taking a full breath was reduced, which was confirmed by spirometric tests that showed a decrease in the difference between FVC (forced vital capacity) before and after exercise with successive compressions. This parameter depends on the size of the lungs, and it seems that in patients who have been infected with SARS-CoV-2 virus, the subjective feeling of having a problem with taking a full breath may be caused by lung weakness and a decrease in lung efficiency and flexibility. Problems with taking a full breath and consequent hypoxia might also explain how HBO works. Hypoxia might be a possible pathogenic mechanism of post-COVID, and in this study, HBOT was used in pathological conditions induced by this negative factor. Related to this, it could be remarked that the mean initial saturation value of patients in this study was 97 (minimum 93) while Cennellotto et al. qualified patients with acute COVID-19 for HBO therapy when their saturation reached 90 [12].

Endurance and strength tests clearly showed a positive effect of hyperbaric oxygen therapy on patient performance, which can also be explained by deconditioning of hypoxia. Before hyperbaric chamber therapy, four patients were unable to continue a two-minute step due to malaise, and after 15 compressions, the average score was 126 ± 38.7 (max 252) steps. The improvement in performance was confirmed by a thirty-second standing test. In addition, there was a significant change in strength (in Newtons) in both the left arm and right arms, as measured with a dynamometer during therapy. There was also an improvement in flexibility, as measured by the sit and reach test, which, along with improved strength, manifested itself in the subjective sensation of being able to take in a full breath of air, and objectively in an improvement in FVC and thus, a reduction in hypoxia, among other things. Venous blood tests showed a reduction in anion gap and lactate levels, thereby reducing metabolic acidosis. It should be emphasized that blood gas analysis is an important element that relates to the patient’s acid–base balance and oxygenation status [26,28,29,30]. It should be noted that the series of exercises and tests used in this study may have been exhausting for patients who had been previously infected with COVID-19. Therefore, it can be concluded that the lack of an appropriate tool to measure fatigue may have been one of the limitations of the work presented. Therefore, a future study could implement a survey that measures patients’ subjective feelings of exhaustion, fatigue, and dyspnea (e.g., the Borg scale).

There may be several reasons for poor prognosis in patients suffering from type 2 diabetes mellitus. Diabetes negatively affects the condition of the complicated alveolar-capillary network of the lungs. Related to this, it should be emphasized that diabetes mellitus causes microvascular damage in patients with lung disease. Therefore, patients with diabetes mellitus often report respiratory symptoms and are at increased risk of several lung diseases. When looking at the molecular mechanisms involved in microcirculatory damage in diabetes mellitus patients, we must mention chronic inflammation. Indeed, in patients with type 2 diabetes mellitus, insulin resistance and altered glucose homeostasis lead to alveolar capillary microangiopathy and interstitial fibrosis through excessive inflammation. Several molecular mechanisms that are induced by excessive inflammation have been suggested to explain microvascular disease and the resulting endothelial dysfunction and lung damage observed in diabetes mellitus patients. One such mechanism of these pro-inflammatory small vessel endothelial pathways involves interleukin 6 (IL-6). IL-6 is a well-known biomarker of inflammation and metabolic dysfunction and has been suggested to be a predictor of lung disease severity. In a study by Sardu et al., a possible association between type 2 diabetes mellitus and COVID-19 infection was found. Indeed, patients with type 2 diabetes mellitus show a high incidence, illness severity, and mortality rate during COVID-19 infection. Furthermore, rates of severe illness are significantly higher in patients with diabetes compared to those without diabetes (34.6% vs. 14.2%; *p* < 0.001) [31]. Diabetes is also a poor predictor of cardiovascular complications in COVID-19 patients. In one study, D’Onofrio et al. [32] demonstrated that within the COVID-19 autopsy cohort, the “diabetes mellitus” vs. “non-diabetes mellitus” specimens had a significantly higher number of SARS-COV-2 particles localized inside the cardiomyocytes. Another study by D’Onofrio et al. shows that the upregulation of ACE2 expression (both total and glycosylated forms) in diabetes mellitus cardiomyocytes, together with an increase in non-enzymatic glycation, may increase susceptibility to COVID-19 infection in diabetes mellitus patients by promoting cellular entry of SARS-CoV2 [32]. An interesting study by Sardu et al. shows that hypertensive COVID-19 patients with an O-negative blood type have significantly higher prothrombotic indices, as well as higher incidences of cardiac injury and death, compared to patients with an O blood type. Furthermore, having an ABO blood type is associated with a worse prognosis in hypertensive patients with COVID-19 infection [33]. An interesting paper by Marfella et al. showed a strong and significant relationship between higher thrombotic viral load and larger thrombus dimension. This is a novel finding that raises the question of the effectiveness of a more aggressive anticoagulant therapy in patients with high cardiovascular risk, such as ASAP patients (asymptomatic SAR-COV-2 positive) with diabetes, hypertension, and dyslipidaemia [34].

The results obtained in this study may lead to the conclusion that HBO may lead to improved gas exchange in patients with complications after COVID-19. In contrast, hyperbaric oxygen therapy was unable to abolish the symptoms of acidosis. It has been well-described that CO2 that is not effectively eliminated by the lungs can enter cells and cause paradoxical intracellular acidification [35]. However, it should be strongly emphasized that this was not the case in our study, as we observed improved gas exchange in our patients. The likely reason is that we are dealing with metabolic acidosis, not respiratory acidosis, the former of which does not depend on gas exchange through the lungs. The explanation for this may be the high anion gap score with low HCO3- parameters. In addition, it should be noted that one of the patients suffered from diabetes mellitus and obesity, both of which may have a negative impact on some metabolic processes. 

Since post-COVID syndrome is a direct consequence of SARS-CoV-2 infection, the course of the initial disease (COVID-19) should also be considered. Thus, it has been noted that patients with severe COVID-19 should be closely monitored for the development of lactate acidosis, acidosis, and decreased renal function [29]. Furthermore, it has been shown that patients with severe COVID-19 are more likely to develop alkalosis than acidosis [30,35,36,37]. In brief, most of the analyzed patients admitted to the intensive care unit showed alkalemia in arterial blood gases, with a higher pH and a lower oxygen partial pressure in arterial blood gas being significantly associated with survival [38]. Thus, the patients presented in our study represent a minority of patients affected by COVID-19, and additionally represent a group of patients particularly vulnerable to the negative consequences of COVID-19. Moreover, the work of the Modena COVID-19 Working Group (MoCo19) showed that acid–base changes were found in 79.7% of patients (metabolic alkalosis 33.6%, respiratory alkalosis 30.3%, combined alkalosis 9.4%), with respiratory acidosis at 3.3%, metabolic acidosis at 2.8%, and other compensated acid–base disorders at 3.6% [37]. Of particular note is the fact that the patients we described were in very severe general condition both during and after the acute infection. This situation may be reflected in the work of MoCo19, since in one study by these authors, all the patients suffering from metabolic acidosis died at the end of follow-up [37].

Another important aspect of the current study was the evaluation of parameters related to neurological/psychological disorders. The results showed a statistically significant improvement in both working memory (*p* = 0.015) and concentration and attention (*p* < 0.001) (Appendix A). These results are indirectly confirmed by the work of Bhayat et al. [14] which demonstrated that HBOT could improve brain perfusion capacity in a 55-year-old Caucasian male.

In addition, it is important to emphasize the extremely important role of vaccination against SARS-CoV-2. In this study, none of the included patients were vaccinated. However, numerous studies unequivocally show the efficacy of vaccination in preventing infection prior to illness or (due to the changing phenotype of the virus) leading to a much milder course of the disease in the case of post-vaccination illness. These observations were verified by Taus et al. [39] in a study of the cellular immune response in convalescent and vaccinated patients.

This paper presents the results of the effect of HBO therapy on the health parameters of unvaccinated patients suffering from post-COVID syndrome. Due to a lack of similar studies in the literature, the authors encountered a serious problem when discussing their results in the context of data presented by other researchers. Nevertheless, a significant improvement in fatigue parameters was observed among patients. Moreover, improvements in cognitive, psychiatric, fatigue, sleep, and pain symptoms were observed among post-COVID patients treated with HBOT [40]. These findings are supported by the literature [13,14].

The observed improvement in neurological parameters was also confirmed by brain perfusion magnetic resonance imaging in a case report by Bhayat et al. [14]. Both the results obtained by Bhayat et al. and those presented in this paper are confirmed by the findings of Zilberman-Itskovich et al. These authors conducted a randomized clinical trial of HBOT and found that the clinical results were associated with significant improvements in brain MRI perfusion and microstructural changes in the supratentorial area, left accessory area, right insula, left precentral frontal cortex, right middle frontal cortex, and superior corneal radius. These results indicate that HBOT can induce neuroplasticity and improve cognitive and psychiatric symptoms, fatigue and sleep problems, and pain in patients suffering from post-COVID-19 conditions. The beneficial effect of HBOT may be attributed to increased brain perfusion and neuroplasticity in regions associated with cognitive and emotional roles [40].

The lack of an identified control group can be considered the most significant limitation of the presented study. In this regard, it is important to note the small number of patients who underwent therapy. In addition, the study scheme basically considered a group of 40 patients. Such a study design was approved by the relevant Bioethics Committee. Once again, it can be emphasized that the purpose of this study was primarily to help patients suffering from long-term symptoms after COVID-19. Therefore, we deliberately chose not to include a control group at this stage of the study. Nevertheless, it should be noted that the presented work describes the preliminary results of HBOT application in patients with post-COVID syndrome, and that the results obtained can be considered promising and may provide a perspective for the implementation of future studies such as randomized clinical trials.

An important aspect of this work, on which much emphasis was placed, was the safety of the qualified patients. Of course, this is an aspect that other researchers have also paid attention to [11,12,27]. For the sake of patient safety, Cannellotto’s team used the minimum dose of HBO (1.45 ATA) in their study [12]. In the present study, the therapeutic dose (2.5 ATA) was used. However, because of this, some patients whose negative symptoms after COVID-19 could indicate the need for HBO did not qualify for therapy. This is undoubtedly a major limitation of HBO therapy, but it should go without saying that patient safety and the potential benefit to them should come first. It is likely that these patients could be offered a minimal therapeutic dose of HBO (as suggested by Cannellotto et al.) [15], but such a recommendation would require that additional appropriate studies be carried out [13].

Based on the results obtained, HBO therapy was temporally associated with an effective treatment outcome in a treated group of patients with post-COVID syndrome. This effect can be attributed, inter alia, to the immunomudulatory properties of HBO therapy [11,12,27]. Here, mention can be made of the effectiveness of HBO in the treatment of asthma [8] and numerous other autoimmune disorders such as atopic dermatitis [9,10]. 

In conclusion, it should be noted that post-COVID syndrome is associated with several negative consequences. Negative symptoms after COVID-19 include poor quality of life; long-term persistent symptoms, including fatigue, shortness of breath, anosmia, cough, sleep disturbances, chest pain, arthralgia; and poorer overall mental health [1,2,3,4,5,6,40]. A comprehensive systematic review and meta-analysis by Malik et al. [41] found that despite the well-established literature on persistent symptoms associated with post-COVID syndrome, the risk factors for their development are still unclear. The authors also noted the important role of pandemic control efforts, which mainly involve through vaccine development. However, given the large population of patients who have recovered from COVID-19 infection, some of whom have developed post-COVID syndrome (leading to, among other things, impaired quality of life), health care attention should focus on understanding the risk factors that trigger post-COVID syndrome and developing appropriate follow-up and treatment strategies [42]. Given this perspective, HBOT appears to be a potentially effective method for improving the negative symptoms of post-COVID syndrome [4,5,6,7,11,12,13,14,20,21,22,23,24,25,26,27,41,42,43].

## 5. Conclusions

A set of 15 compressions at HBO coincided with significant and sustained improvement in quality of life, endurance and strength, specific gasometric and spirometric parameters, and working memory and attention. 

Considering the results obtained, including the apparent improvement in the clinical condition of the patients, it can be concluded that the application of 15 compression sessions can lead to noticeable improvements in health and performance parameters as well as improvements in specific blood gas parameters. Thus, the proposed therapy can be recommended for use in randomized clinical trials in patients suffering from the persistent negative effects of post-COVID syndrome after SARS-CoV-2 infection.

## Figures and Tables

**Figure 1 jcm-12-00308-f001:**
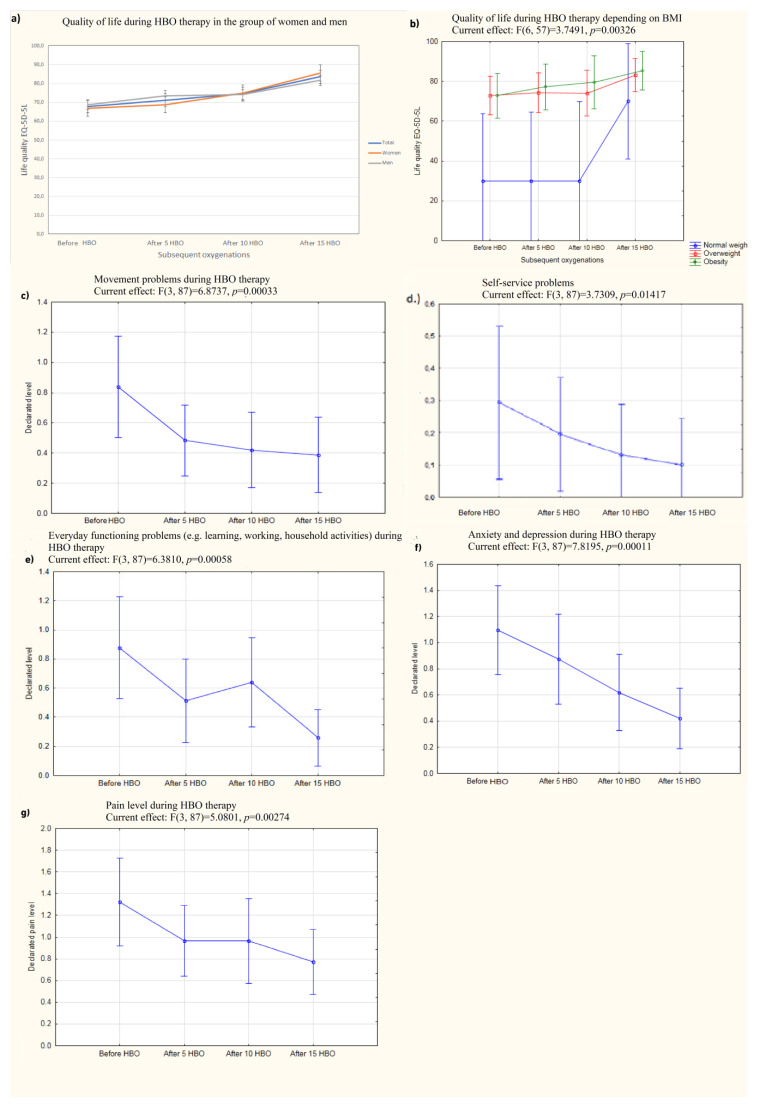
The quality of life level depending on the selected parameters. (**a**) Change in patients’ quality of life during hyperbaric oxygen therapy in a group of women and men; (**b**) change in quality of life depending on BMI; (**c**) change in the declared mobility problems during the course of HBO therapy; (**d**) change in the declared problems with self-service during the course of HBO therapy; (**e**) change in the declared problems with performing daily activities during HBO therapy; (**f**) change in the sense of anxiety or depression during HBO therapy; (**g**) change in pain experienced during HBO therapy.

**Figure 2 jcm-12-00308-f002:**
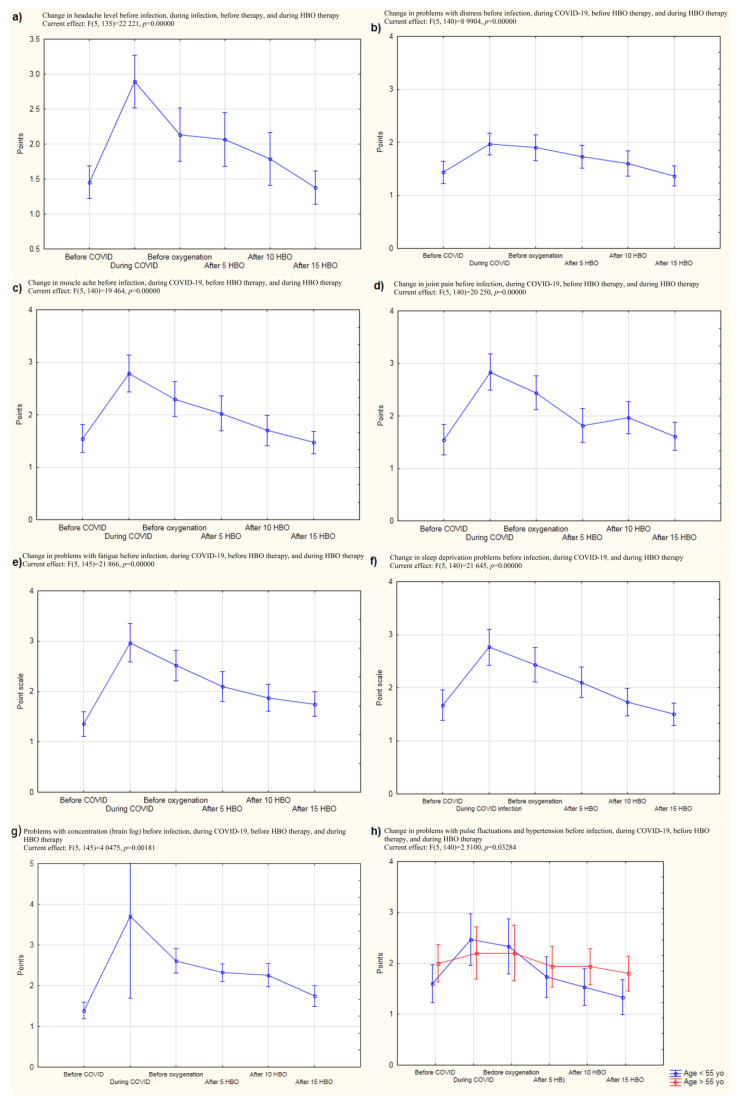
The results of the subjective assessment of symptoms characteristic of COVID-19 and post-COVID before the infection, during the infection, before the experiment, and after 5, 10, and 15 HBO compressions. (**a**) Change in headache level before infection, during infection, before therapy, and during HBO therapy; (**b**) Change in problems with distress before infection, during COVID-19, before HBO therapy, and during HBO therapy; (**c**) Change in muscle ache before infection, during COVID-19, before HBO therapy, and during HBO therapy; (**d**) Change in joint pain before infection, during COVID-19, before HBO therapy, and during HBO therapy; (**e**) Change in problems with fatigue before infection, during COVID-19, before HBO therapy, and during HBO therapy; (**f**) Change in sleep deprivation problems before infection, during COVID-19, and during HBO therapy; (**g**) Problems with concentration (brain fog) before infection, during COVID-19, before HBO therapy, and during HBO therapy; (**h**) Change in problems with pulse fluctuations and hypertension before infection, during COVID-19, before HBO therapy, and during HBO therapy.

**Table 1 jcm-12-00308-t001:** Change in quality of life parameters during hyperbaric oxygen therapy.

Measured Parameter	Before HBO	After 5 HBO	After 10 HBO	After 15 HBO	Significant Improvement YES/NO
Quality of life (µ ± SD):					YES (*p* < 0.001).
All Patients	67.8 ± 18.5	71.2 ± 18.5	74.6 ± 19.1	83.6 ± 13.6	
Men	68.8 ± 18.1	68.7 ± 20.0	75.0 ± 19.0	85.7 ± 10.9	
Woman	66.8 ± 19.0	73.6 ± 16.7	74.3 ± 19.2	81.7 ± 15.5	
Walking problems (studied population %)					YES (*p* < 0.01)
no problems	45%	58%	68%	71%	
slight problems	29%	35%	23%	19%	
moderate problems	23%	6%	10%	10%	
severe problems	3%	0%	0%	0%	
Self-service problems (studied population %)					YES (*p* < 0.01)
no problems	77%	84%	90%	94%	
slight problems	19%	13%	6%	3%	
moderate problems	0%	3%	3%	3%	
severe problems	3%	0%	0%	0%	
Daily activities (studied population %)					YES (*p* < 0.01)
no problems	42%	61%	55%	77%	
slight problems	39%	29%	29%	19%	
moderate problems	10%	6%	13%	3%	
severe problems	10%	3%	3%	0%	
Sense of anxiety or depression (studied population %)					YES (*p* < 0.01)
no problems	29%	42%	58%	65%	
slight problems	39%	35%	23%	29%	
moderate problems	26%	16%	19%	6%	
severe problems	6%	6%	0%	0%	
Change in the intensity of pain (studied population %)					YES (*p* = 0.027)
no problems	29%	35%	42%	45%	
slight problems	26%	35%	32%	32%	
moderate problems	29%	26%	13%	23%	
severe problems	16%	3%	13%	0%	

## Data Availability

The data presented in this study are available on request from the corresponding author.

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
