# Peer review of "Preliminary Research on the Effect of Hyperbaric Oxygen Therapy in Patients with Post-COVID-19 Syndrome"

_jcm, 2022, doi:10.3390/jcm12010308_

Round 1

Reviewer 1 Report

This is overall a very interesting and important prospective study about HBOT in PCS, and with 31 patients of reasonable size of the included cohort.

The inclusion criteria were clear, and the numerous clinical tests throughout the HBOT give a good impression about the clinical course of the therapy and the final outcome weeks after completion of the HBOT.

However, and this need to be emphasized in a limitations-section, there was no control group without HBOT, neither a sham-exposure. Thus, the improvement of symptoms, although impressive, cannot be verified against controls or sham. This will be a problem in a number of similar pilot studies before performing complete RCTs (as meanwhile the first published from Israel), but nevertheless needs to be mentioned.

Another critical point are the subjective self-assessments (fig. 2), here particular before and during COVID. Retrospective assessments are always difficult, but in this study COVID-infection was in mean over 6 months ago. Thus, a reputable assessment after this period should be critical discussed by the authors or even left out from results and figures.

In this context, the fig. 2 should be modified: if the authors decide to leave the retrospective assessments in the figures a-h, then I would recommend to make a gap between “during COVID” and HBOT. The time here was about 6 months (mean), and the HBOT overall 3 weeks. Otherwise the reader has the impression of spontaneous improvement of symptoms regardless HBOT.

Minor: all ordinates should begin at zero, 2a) too.

Fig 1: why is a) plotted as bar graph and not b) as well? Both are about EQ-5D-5L and should be plotted similar and always with error bars. 

Why was the quality of life of the normal weight participants initially so low? This surprising fact should be discussed – was it because of the small number of patients with normal weight? From line 218 I conclude that only 3 patients were of normal weight? Also, a statistical test with large differing cohort sizes is difficult and should be critically discussed. Maybe overall, it is better to leave the weight discussion out and to delete fig. 1b.

Discussion: ll 421ff: in this paragraph the authors discuss their bloodgas results from HBOT 6 months after COVID suffering from PCS and compare their data with cases of acute and severe COVID in hospital. In fact, I cannot follow the argumentation in this paragraph, and I don´t see the direct connection between hospital stay of patients from literature and the actual bloodgas-data from HBOT in PCS, 6 months later. This should be clarified. I do understand, that obese and moreover diabetic patients may present with altered bloodgases before HBOT and maybe elevated lactate, but the discussion should better focus solely on the HBOT period in this study.

Minor: throughout the manuscript language could be improved and a number of typing errors and missing spacings should be corrected.

Author Response

Dear Reviewer, thank you for your positive review of our work and a number of valuable comments. We would like to thank you so much for insightful work with this manuscript. We try to improve all the parts of the manuscript according your valuable comments and suggestions. There is no doubt that your comments and remarks will improve the quality of entire manuscript.

Reviewer 2 Report

The aim of this manuscript is to provide evidence for the effectiveness of HBOT for post covid condition. It shows a temporal association with 15 sessions of HBOT and improvement of objective tests, symptoms and self reported HRQoL; information that can be valuable for designing clinical trials.  

Despite good intentions, the study design and methodology does not support the aim of the study. First of all without a control group no conclusions can be drawn regarding efficacy, there is merely a temporal association. Secondly, an interventional study involving medical oxygen is generally regarded as a clinical trial and should be conducted in compliance with ICH-GCP and registered with EMA according to European legislation. If the treatment has a waiver from this praxis, due to national legislation in Poland, this should be described in the manuscript.

https://www.globalcompliancenews.com/2021/06/11/poland-new-draft-act-on-clinical-trials-of-medicinal-products-for-human-use17052021/

https://www.traple.pl/en/new-proposal-for-a-polish-act-on-clinical-trials/

There has also been a number of new publications since your manuscript was written/submitted. There is now international consensus regarding the diagnose post covid condition (U09.9) and the incidence is regularly changed. A recent systematic review and meta-analysis has been published (Malik 2022). More importantly an RCT (Zilberman-Itskovich 2022) has been published, suggesting effect of 40 sessions in a double-blinded, placebo-controlled trial and another RCT, also double-blinded, placebo-controlled but with only 10 sessions has a published protocol and is well under way (Kjellberg 2022).

I would suggest that you discuss your trial/manuscript with other senior clinical researchers and maybe the Bioethics Committee of the Medical University of Silesia in Katowice, Poland that approved the trial. You have some important findings and it would be a waste of effort not to publish the data.

Author Response

(The authors gave the same response as above.)

Reviewer 3 Report

In the introduction, there is no clearly stated aim(s) and hypothesis of the study, which should definitely be added. Furthermore, I think the title should be changed to be more specific and related to the study itself, perhaps: “The hyperbaric oxygen therapy and quality of life in patients with post-COVID-19 syndrome”

In the materials and methods chapter, a brief description of the spirometer used and the measurements made with an explanation of the abbreviations used later in the results should be added.

The manuscript is clear, presented in a well-structured manner, and relevant to the field.

Cited references are recent and relevant.

Results seem highly reproducible based on the details given.

Conclusions are consistent, clear, and supported by the results and all possible limitations are clearly pointed out.

Regarding inaccuracies within the text, there are some, including typos and abbreviations explanations missing.

Overall, research has its own scientific and clinical value and could be considered for publication after minor changes.

Author Response

(The authors gave the same response as above.)

Round 2

Reviewer 2 Report

Dear Authors,

You have some important and very interesting data that I believe is worth publishing but the quality of evidence can never be better than the design of the study. To my understanding this is a Case series (uncontrolled longitudinal study). A series of individuals (n=31) have received an off-label treatment, in this case 15 sessions of HBOT and have been thoroughly evaluated by a number of objective tests and self-reported questionnaires before and after the treatment (before/after design). There is a temporal association with improvement in both self-reported and objective finding and a causal relationship can be discussed/suggested considering possible confounding factors. Since this is a novel indication, I strongly suggest that you report any adverse events if you have recorded them, for credibility. I hope my comments and suggestions below will improve your manuscript.

Sincerely, 

Abstract:

Line 32-34 An uncontrolled longitudinal study design cannot prove that an association reflects cause and effect. Please revise. Suggested change: “may bring about” to  “were temporally associated with” or “coincided with”

Introduction:

Line 62-67 The randomized trials had their protocols published on ClinicalTrials.gov in late 2020 “Hyperbaric Oxygen Therapy for Post-COVID-19 Syndrome (HBOTpCOVID)” and early 2021 “Safety and Efficacy of Hyperbaric Oxygen Therapy for Long COVID Syndrome (HOT-LoCO)”. Please revise to “past tense” and explain that this was the evidence available when you started your study.

Line 70-76 The previous case-reports may be mentioned, in that case in “past tense” but are of limited interest when an RCT is published. I suggest you add that to the introduction.

Materials and methods:

It can be argued that this is not a clinical trial but it should at least include when ethical approval was approved, between what dates subjects were included and how many patients were screened, excluded and reason for exclusion. If it is regarded as a non-clinical trial there is no need for pre-registration but it is recommended to register protocols for all interventional studies. I suggest that you tone down the “interventional” part and re-write as a case-series with follow up.

2.4 Long term follow up, it can be debated if 2 weeks is a long term follow-up, normally long term follow up is at least 6-12 months for hyperbaric oxygen.

Results: Since the data is exploratory in its character, it would be easier to digest if you try to limit the results, they are interesting enough without reporting every detail. 

3.1 EQ-5D

It is not possible to compare health profiles between individuals and the individual values can not be used for assessing treatment effect. To compare individuals or association with an intervention the “index” should be used. The results show reports the HRQoL “today” and cannot be used for retrospective assessment, please remove the “Before COVID” result in figures. Please refer to https://euroqol.org/eq-5d-instruments/ where you can find instructions. Please revise.

I also do not understand the figures what is on the y-axis? EQ-5D is an ordinal scale that creates categorical variables 1-5. It looks like the graphs in Figure 1 show mean and SD. Please revise.

As above with your own questionnaire, it is categorical variables, seemingly ordinal but looks like mean and SD is presented in Figure 2. Please revise and/or explain.

3.2 Assessment of general physical condition, including efficiency

Line 279-332 I would suggest that you remove the concluding remarks regarding effect caused by Hyperbaric oxygen, again due to the study design and also please read my remark regarding GET as a possible confounder in this patient group.  

Discussion:

Line 530-531 Same as in abstract. An uncontrolled longitudinal study design cannot prove that an association reflects cause and effect. Please revise. Suggested change: “may bring about” to  “were temporally associated with” or “coincided with”. I suggest you write the lack of a control group as the first and most important limitation.

Line 535-548 I suggest you re-write as suggested and move this to “introduction”

There were a large number of exercise interventions during the observed time, in Supplement 1 you show on the change in pulse that the exercice was quite exhausting, ideally Exhaustion/Fatigue/Dyspnea on Borg-scale would also be reported. For patients with post covid condition this may have had impact on the outcome, as a matter of fact it can be argued that you have also performed a Graded exercise test (GET) over 4 weeks, which has been suggested as a therapeutic intervention. This should be discussed as a confounding factor that may have contributed to the positive change over time, again only an association.

Conclusions:

Line 550-558 Same as in abstract and discussion. An uncontrolled longitudinal study design cannot prove that an association reflects cause and effect. Please revise.

Sincerely, 

Author Response

Dear Reviewer,

Thank you for the positive review of our manuscript. All your valuable remarks have been implemented. Please see the attachment

Yours faithfully,

Wojciech Łabuś
